# Real-world clinical practice of Diabetic Foot Ulcer prevention and care in Singapore: A qualitative inquiry with healthcare professionals

Anita Pienkowska[1], Josip Car[1,2‡], James Best[1‡], Choon Huat Gerald Koh[3‡], Lorainne Tudor Car[1‡], Kavita Venkataraman[3‡], Bernhard O. Boehm[4‡], Huiling Liew[4‡], Elaine Tan[5‡], Harvinder R.S. Sidhu[6‡], Tavintharan Subramaniam[7‡], Rosalind Siah Chiew Jiat[8‡], Naren K. Surendra[1‡], Kelley Fann Ing Goh[1‡], Nandika Lodh[1‡], Andy Hau Yan Ho[1,9‡*]

1 Lee Kong Chian School of Medicine, Nanyang Technological University, Singapore, Singapore, 2 School of Life Course & Population Sciences, King's College London, London, United Kingdom, 3 Saw Swee Hock School of Public Health, National University of Singapore, Singapore, Singapore, 4 Department of Endocrinology, Tan Tock Seng Hospital, Singapore, Singapore, 5 National Healthcare Group Polyclinics, Singapore, Singapore, 6 National University Health System, Singapore, Singapore, 7 Khoo Teck Puat Hospital, Singapore, Singapore, 8 National University of Singapore, Singapore, Singapore, 9 School of Social Science, Nanyang Technological University, Singapore, Singapore

☯ These authors contributed equally to this work.
‡ JC, JB, CHGK, LTC, KV, BOB, LH, ET, HRSS, TS, RSCJ, NKS, KFIG,NL, AHYH also contributed equally to this work
* andyhyho@ntu.edu.sg

## Abstract

### Aim

People living with diabetes are at risk of developing diabetic foot ulcers (DFU). While international and local clinical care guidelines and pathways have been formulated to optimize the prevention and treatment of DFUs, a continuous audit of real-world adherence among healthcare professionals (HCPs) is needed to ensure care quality, safety, and efficacy.

### Methods

A qualitative study design involving focus group discussions was used to explore practices in the prevention and treatment of DFUs. Verbatim transcripts from eight discussions involving 19 HCPs, purposively sampled, were analyzed using the frame-work method. Coding was guided by a DFU model of care developed from international and local clinical guidelines.

### Results

Clinical practices for DFU prevention and care management in Singapore generally adhere to existing guidelines, though risk stratification and DFU classification are not commonly performed. During clinical visits, HCPs perform foot assessments that

**Data availability statement:** Deidentified interview data are available at NTU Data repository: Center for Population Health Sciences, 2025, "Real-world clinical practice of Diabetic Foot Ulcer prevention and care management in Singapore: A qualitative comparative inquiry with healthcare professionals", https://doi.org/10.21979/N9/NDYXRX, DR-NTU (Data), V1

**Funding:** This study was funded by the Singapore Ministry of Health's National Medical Research Council under the Health Services Research Grant (HSRG-DB17Nov002), awarded to JC and AHYH. The funder nad no role in the study design, data collection and analysis.

**Competing interests:** The authors have declared that no competing interests exist.

**Abbreviations:** ABPI, Ankle-Brachial Pressure Index, ACE, Agency for Care Effectiveness, DEFINITE, Diabetic Foot in Primary and Tertiary (Care), DFU, Diabetic Foot Ulcers, DNE, Diabetic Nurse Educator, FGD, Focus Group Discussion, HCP, Healthcare Professional, HDB, Housing Development Board, IWGDF, International Working Group on Diabetic Foot, LEA, Lower Extremity Amputation, LEAPP, Lower Extremity Amputation Prevention Programme

encompass mainly visual inspection, evaluation of vascular status and neurological status. Education on DFU prevention and management is extensive across all diabetes care. Referrals to podiatrists include cases beyond active wounds and high-risk issues.

## Conclusion

Implications for practice are considered and highlight the need for a clearer delineation of roles among HCPs in DFU clinical care guidelines. This study provides a guide for further studies in the area of patient management.

## Introduction

Diabetes mellitus is a prevalent chronic condition affecting millions worldwide and has emerged as a pressing public health concern [1]. People living with diabetes are susceptible to the development of diabetic foot ulcers (DFU), a serious complication that can lead to lower extremity amputation (LEA) or even premature mortality if not managed effectively [2]. It is estimated that 19% − 34% of people living with diabetes are at risk of developing DFU within their lifetime, and that 20% of these individuals may progress to require LEA, be it minor (below the ankle), major (above the ankle), or both [2,3]. Similarly in Singapore, DFU and LEA posed significant clinical and economic challenges [4].

With the prevalence of diabetes in Singapore projected to escalate from 7.3% in 1990 to 15% by 2050 [5], targeted efforts have been implemented to enhance DFU prevention and management. Building on the foundational guidelines established by the International Working Group on the Diabetic Foot (IWGDF) [6–8], an international consensus specific to the Asia-Pacific region was subsequently developed [9]. In the context of Singapore, the Agency for Care Effectiveness (ACE) has introduced a foot assessment guide [10] comprising elements of proper foot assessment, referral pathway, as well as patient education principles and topics. Moreover, within the local context, initiatives like the Lower Extremity Amputation Prevention Program (LEAPP) and Diabetic Foot in Primary and Tertiary (DEFINITE) Care programs have been established to streamline the process of DFU care following multidisciplinary team approach [11–13]. The efforts resulted in significant reduction in minor and major amputation rates [14].

While the multidisciplinary team approach yielded improvements in DFU prevention and treatment, it is not the sole method employed within the healthcare system in Singapore. A detailed audit of real-world practices is crucial to uncover the key drivers of success and pinpoint gaps or overlaps that may limit further progress. The primary objective of this study is to analyze actual clinical practices for DFU care in Singapore and to compare them with international and local guidelines. Unlike studies that focus solely on patient adherence [15] or comparisons between national and international guidelines [16], this research offers an examination of the alignment between real-world practices and established standards. This study aims to provide

healthcare policymakers and care pathway architects with critical insights into overlaps and gaps in DFU prevention and treatment.

## Materials and methods

### Study design

This constructivist qualitative study employed focus group discussions (FGD) to elicit information on DFU prevention and care practices in Singapore. As the purpose of this study was to examine DFU prevention and care management practices and juxtapose them with the clinical guidelines, the framework method [17] was adopted to analyze and compare qualitative data with specific clinical guideline items to uncover overlaps or gaps in service provision. This study followed the Standards for reporting qualitative research (SRQR) [18] (S1 Appendix).

### Sampling

Participant recruitment was conducted at five major public hospitals and medical clinics in Singapore covering two out of three healthcare clusters. The initial study design included all three clusters to capture a broad spectrum of service settings and ensure diverse perspectives on DFU prevention and treatment. However, one cluster opted not to participate further due to differing institutional policies related to data sharing considerations for a connected sub-project. Potential study participants were identified with purposive sampling by site coordinators to ensure a diverse array of healthcare disciplines and a wide range of experience. The inclusion criteria comprised healthcare professionals (HCP) involved in DFU care.

### Ethical approval

This study was approved by the National Healthcare Group Domain Specific Review Board (NHG: DSRB Ref: 2021/00618 dd. 7.02.2022). The recruitment period for this study was from 23rd March to the 31st August 2022. The participants provided written consent.

### Data collection

Twenty-one HCPs expressed interest in participating in the study, 20 signed informed consent, and 19 participants completed a demographic information sheet and attended FGDs. No dropouts occurred. Data was collected from eight FGDs through the Zoom (Zoom Video Communications, Inc.), each lasted about 60 minutes and was guided by an interview guide informed by a literature review (S2 Appendix). They were informed about audio recording and assured of confidentiality. FGDs were led by a senior clinician-researcher and senior social scientist with expertise in mixed-methods healthcare research who had no prior relationship with the participants (JC and AHYH; both male). Preliminary analysis of the verbatim transcripts was conducted concurrently with data collection. Based on emerging patterns and the recurrence of themes, JC and AHYH, through iterative discussions, jointly determined that data saturation had been reached after interviewing 19 participants. Preliminary analysis of the verbatim transcripts was conducted during the data collection phase.

### Data analysis

We used the framework method [17] to analyze the data. This method, which originated in social policy research, includes the following steps: transcription, data familiarization, coding, developing analytical framework, indexing by applying the analytical framework, charting data into the framework matrix, and interpretation. The analytical approach was twofold comprising analysis of international and local guidelines to serve as a model of care, and coding of FGD transcripts guided by the DFU care model.

The research team (NKS and AP) developed a high-level model of care, outlining essential components visualized in Fig 1. Key guidelines such as IWGDF [6–8], Diabetic Foot Ulcer Care in the Asia-Pacific region International Consensus Document [9] and ACE [10], were iteratively analyzed to glean recurring and emphasized elements of DFU prevention and care. We began with iteratively drafting a decision flow diagram representing patient journey through different health-care professionals, based on outcomes from various assessment, investigations and treatment strategies criteria (e.g.,

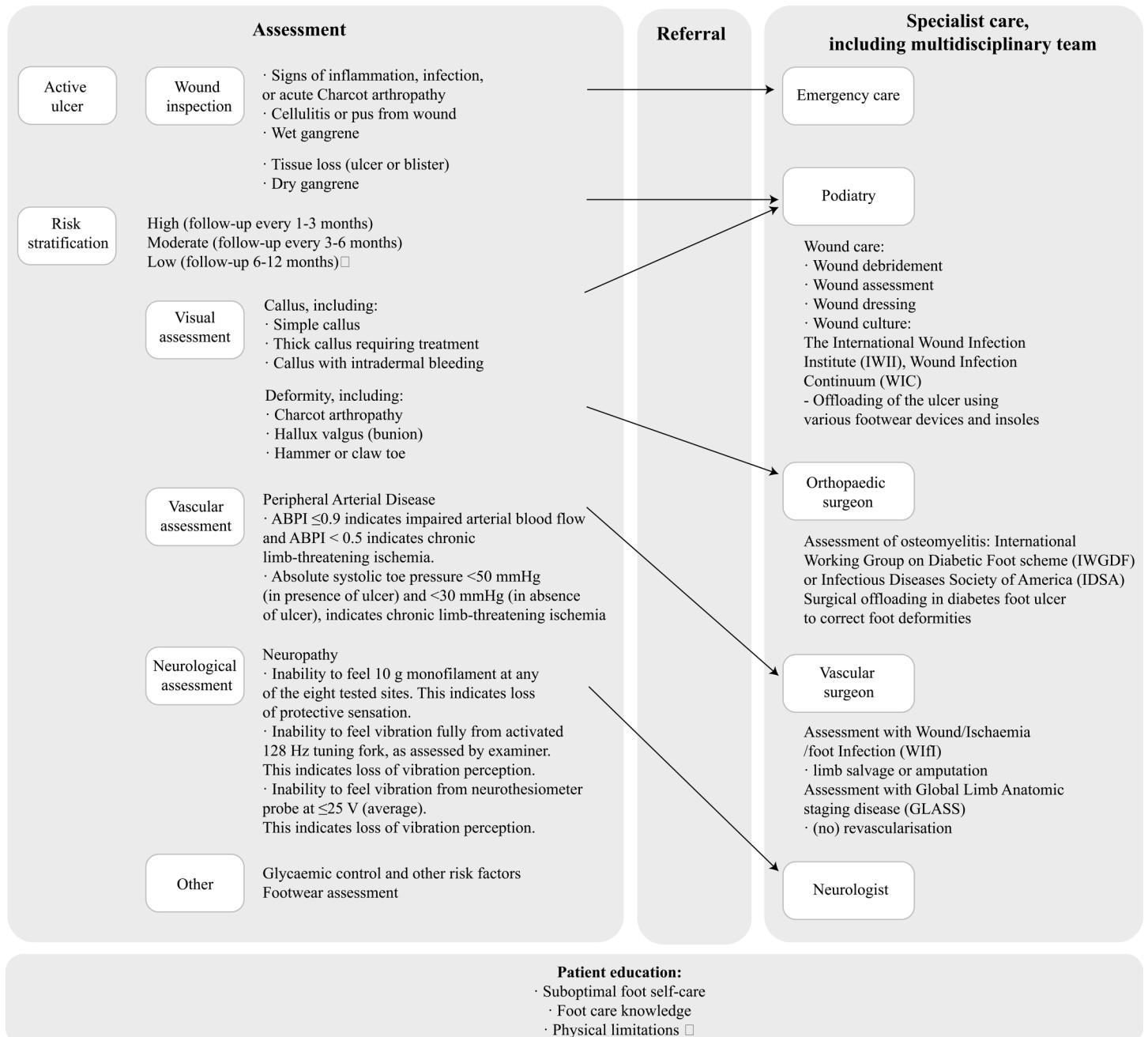

**Fig 1. Model of DFU care.**

presence or absence of an active ulcer) drawn from the guidelines. To enhance clarity and simplify the diagram, we examined the main sections of each guideline to identify priority areas. The ACE guideline emphasizes foot assessment (including active DFU presentation, risk stratification, other contributing factors, and referrals) and patient education. The IWGDF guidelines cover pathophysiology, prevention, assessment and treatment, active Charcot neuro-osteoarthropathy, and organization of care. The Asia-Pacific consensus document highlights assessment, investigations, infection management, clinical treatment, and strategies for improving outcomes, such as multidisciplinary care, patient-centered approaches, self-care, telemedicine, and caregiver involvement. Based on this synthesis, we grouped the previously mapped steps into three interdependent domains—assessment, referral, and specialist care—with patient education as a cross-cutting element.

The team got familiarized with the deidentified transcripts by carrying out a preliminary line-by-line coding process (NL and KG) and by discussing the preliminary codes (NL, KG, AP, AHYH). This was followed by code recategorization, data re-coding (AP) and resolving any discrepancies in categories and codes (AP and AHYH). The data were indexed using Lumivero NVivo (QSR International) and the categories and codes were mapped using draw.io (JGraph Ltd).

The high-level model of care (Fig. 1) guided our iterative coding process. Insights from the FGD transcripts influenced its structure, resulting in a codebook which organizes data on "specialist care" under two themes: wound care and guidelines and offers more nuanced codes for patient education and referrals – the diverse information provided by participants impacted the granularity of codes for educational formats and topics, as well as additional codes for organizing care. In addition, we created a distinct theme for guideline use and wound care practices across specialties (Fig 2) to capture the declared adherence to clinical guidelines and the range of therapeutic and treatment strategies.

## Results

Of the 19 participants recruited, the majority were ethnic Chinese females with more than six years of experience in DFU care (Table 1). The analysis generated five themes including assessment, patient education, guidelines, referrals, and wound care (Fig 2 presents the overview of themes, categories and codes; S3 Appendix presents the codes, frequencies and sample quotes, S4 Appendix presents the distribution of categories and codes among the study participants).

### Assessment: consistency and limitations in screening practice

The study participants provided insights into the various types of foot assessments conducted during consultations. Fig 3 illustrates assessment types and suggests that not conducting any footcare assessment is a rare occurrence among the study participants (2/19 participants), but it is similarly infrequent that the assessment comprises all recommended elements.

According to the study participants, visual inspection plays a crucial role in the assessment and serves as the entry point of the foot screening. HCPs assess the condition of the nails and skin, searching for any signs of potential or active injury, dryness, discoloration, calluses, fungal infection between the toes or nail bed infections. Due to time constraints, the inspection may be scaled down to patient's medical history. Limitations of visual inspection are highlighted by some participants as they pointed out that differentiating asymptomatic cases from the ones on the verge of being symptomatic is challenging and should be of special interest to HCPs.

*"A lot of the times, patients are actually asymptomatic, and I think the challenge here is really differentiating those that is asymptomatic and on the verge of becoming symptomatic. I think this group of patients are the most tricky to deal with."* (S01P04, Senior Podiatrist)

Vascular assessment is a common practice in consultations. This assessment can encompass a variety of techniques, ranging from straightforward palpation of pulses in the lower extremities, such as the dorsalis pedis and posterior tibial

pulses, to more precise methods like the ankle-brachial pressure index (ABPI), toe pressure or (though not included in 2019 ACE guidelines) the Doppler waveform study. Notably, in the case of low vascular status of patients struggling with persistent challenges in wound healing, a podiatrist involves a vascular surgeon to assess the need for revascularization to enable wound healing.

Neurological assessment commonly encompasses a range of symptom evaluations, such as querying the patient about the presence of numbness or reduced sensation of the foot and administering tests like the monofilament test or neurothesiometer for loss of protective sensation and vibration perception testing respectively. One study participant noted that a comprehensive podiatric assessment should ideally include monofilament testing at 12 distinct locations on

| Assessment | Referrals | Wound care | Patient Education |
|---|---|---|---|
| Visual assessment | From | Wound treatment | Formats |
| Vascular assessment | Inpatient setting Primary care Doctor Podiatry Specialist | Prescription | Leaflets One-on-One Learning reinforcement |
| Neurological assessment | | **Guidelines** | |
| Wound inspection | To | Risk stratification or DFU classification | Topics |
| Footwear | Podiatrist Specialist Social worker General Practitioner Psychologist Primary care Nurse Inpatient care | Specific guidelines or classifications | Footwear General diabetes Wound care Importance of inspection Hygiene Trimming nails Moisturising |
| Foot shape | | ACE IWGDF Other | |
| Skin temperature | Working together | Other aspects | Other aspects |
| Other | Post-referral appointments | No guidelines used Unclear literature | Time needed/spent Message tailoring Caregivers |

**Fig 2. Concept map of the categories and codes.**

**Table 1. Sociodemographic information.**

| Participant ID | Role | Gender | Experience in DFU care | Ethnicity |
|---|---|---|---|---|
| S01P01 | Senior podiatrist | Female | 12 Years | Malay |
| S01P02 | Wound nurse | Female | Less than 1 Year | Chinese |
| S01P03 | Advanced Practitioner Nurse in Diabetes | Female | 10 Years | Chinese |
| S01P04 | Senior podiatrist | Female | 11 Years | Chinese |
| S01P05 | Diabetologist | Female | 6 Years | Chinese |
| S01P06 | Vascular Surgeon | Male | 4 Years | Chinese |
| S02P01 | Senior Podiatrist | Female | 10 years | Malay |
| S02P02 | Senior Podiatry Assistant | Female | 10 Years | Malay |
| S02P03 | Advanced Practice Nurse | Female | 12 Years | Indian |
| S02P04 | Diabetes Nurse Educator | Female | 1 Year | Malay |
| S02P05 | Endocrinologist | Male | 2 Years | Chinese |
| S02P06 | Family Physician | Male | 3 Years | Chinese |
| S03P01 | Vascular nurse | Female | 3 Years | Chinese |
| S03P02 | Vascular nurse | Female | 3 Years | Chinese |
| S03P03 | Vascular Surgeon | Male | 7 Years | Indian |
| S03P04 | Nephrologist | Female | 6 Years | Chinese |
| S04P01 | Podiatrist | Male | 13 Years | Chinese |
| S04P03 | Endocrinologist | Male | 15 Years | Indian |
| S04P04 | Vascular Surgeon | Male | 8 Years | Chinese |

the foot, however, in practice, these assessments often focus solely on the toes. In some case, participants resort to a tactile examination instead of using a monofilament, as they don't routinely carry one. To address this challenge IWGDF recommends the Ipswich Touch test. None of the study participants mentioned using a tuning fork, recommended in ACE guidelines.

Footwear assessment revolves around a visual inspection of the shoes worn during the appointment focusing on key aspects such as whether the shoes are open-toe or enclosed, if they appear worn or new, if they fit appropriately, and whether they possess any sharp edges. Additionally, observing the patient's gait while walking can provide valuable information during this assessment. The guidelines also highlight the need for firm heel counter, fasteners such as shoelaces and inner cushioning which were not mentioned by the study participants.

Wound inspection and progression evaluation involves the examination for any discharges on the dressing, the presence of open wounds, or signs of active infection. It also entails an assessment of the wound's healing trajectory and evaluating the effectiveness of the products used in treatment.

HCPs also conduct a basic examination of foot shape, which helps to identify issues like bunions, flat feet, or nail deformities, all of which can lead to friction-related problems within footwear. Notably, none of the participants mentioned looking for hammer toes and hallux valgus, despite them being indicated in ACE guidelines. Moreover, given its emergency status, early identification of acute Charcot foot, which includes looking for any temperature changes, was also considered important.

## Patient education: high value but varied depth

Foot care education has been emphasized repetitively by our study participants as one of the most crucial aspects of a consultation. Only 3 of 19 participants did not mention it at all. The prevailing formats (S4 Appendix) primarily included leaflets and one-on-one meetings. Less commonly used were online resources, videos, and group sessions, such as life

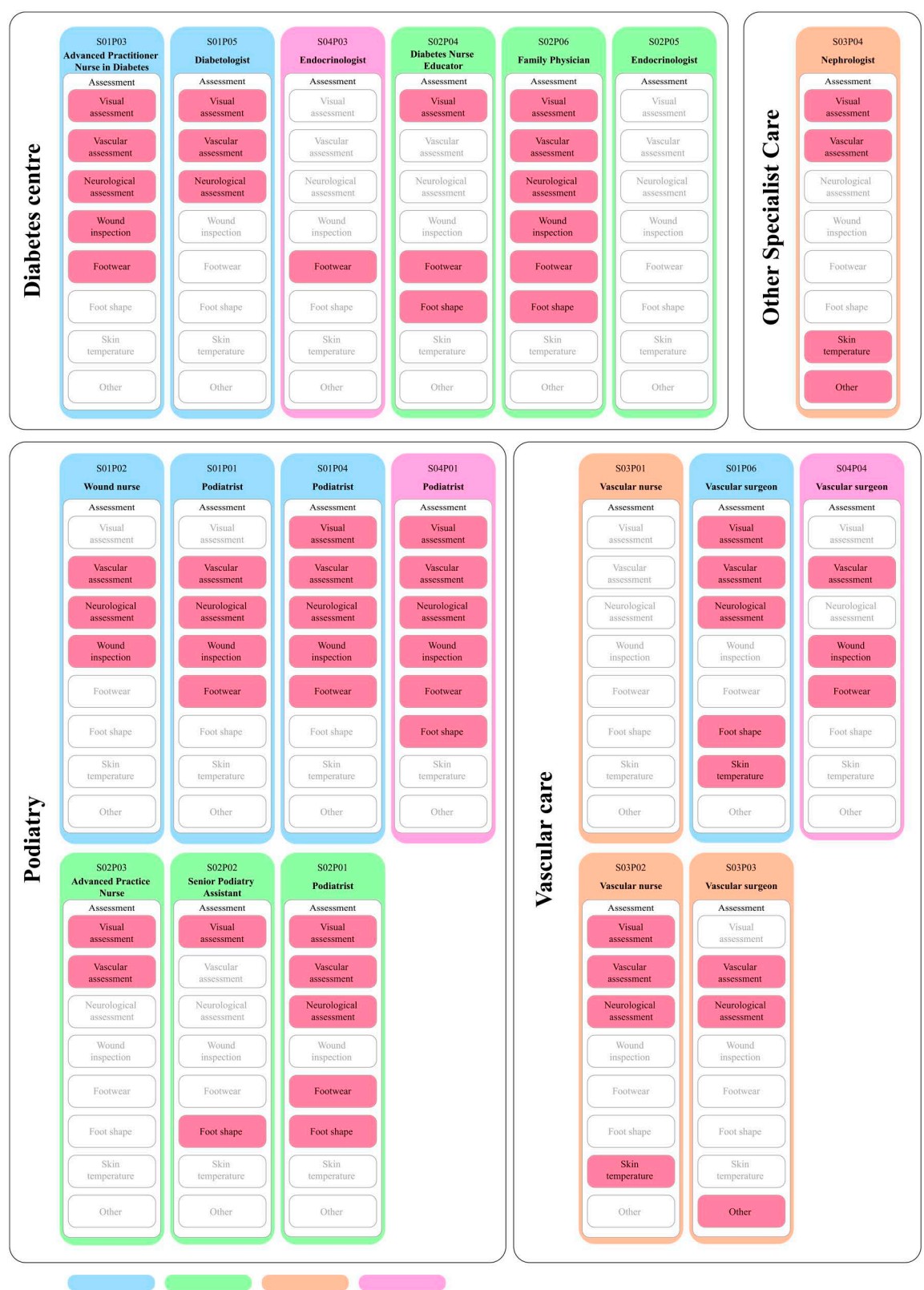

**Fig 3. Visual representation of the codes under "Assessment" category.**

skills programmes. Notably, HCPs use additional strategies in patient education such as reinforced learning, including testing patients' knowledge with questions about previous visits or materials, adapting leaflets to the condition severity or, less common, weekly calls to remind patients about footcare.

Footcare education provided by the study participants primarily focused on topics such as footwear (9/19 participants), including shoe shape and size or the use of socks; general diabetes education (8/19); wounds (8/19); inspection (4/19); hygiene (3/19); nails (3/19); and moisturizing (2/19). However, several specific recommendations outlined in clinical guidelines—such as inspecting shoes before wearing them, ensuring sole flexibility, using breathable materials, adjustable fastenings, or low heels—were not mentioned. This may suggest that these elements are communicated in a different format (i.e., leaflets), hence, not mentioned by the interviewees, or that they are not perceived as immediately relevant in the context of this patient population. Notably, we observed inconsistent recommendations about Crocs™, alternating between endorsing and advising against their use.

> *"You ask them to bring the usual footwear that they usually wear out. So more often than not, it's like some Crocs or slippers, which is completely not suitable…"* (S4P04, Vascular surgeon)

> *"Normally we will advise them not to have open toes shoes, right? So, we will tell them what are the options… like, Crocs shoe, although not escalator friendly lah* [In Singlish, a particle used at the end of sentences or phrases to add emphasis]*, but still okay."* (S03P02, Vascular nurse)

## Guidelines: selective application and gaps in utility

The most common clinical guidelines referred to by study participants include the IWGDF and ACE. Other, less frequently mentioned guidelines include the WIfI (Wound, Ischemia, and foot Infection) Classification and King's Classification for DFU, Transatlantic Inter-Society Consensus for vascular stratification, and general diabetes guidelines.

The majority of study participants did not engage in risk stratification or DFU classification and did not rely on such guidelines (S4 Appendix). Nurses focused on flagging issues for doctors rather than following the risk stratification or DFU classification. Some surgeons rarely if at all made use of clinical classifications, as it was believed that they do not help to inform their work.

> *"Not useful for me. The moment they have a wound, means I have to evaluate and treat. It doesn't matter what's the classification."* (S04P04, vascular surgeon)

One participant, a family physician, introduced a pilot program for stratifying patients. This streamlined screening, increased foot health awareness, and helped identify high-risk patients for podiatrist referral, allowing podiatrists to focus on more urgent cases.

> *"Rather than creating the additional step of sending patients to the podiatrist and podiatrist assistance, what I do for my own patients is that I stratify them according to their risk level. […] I felt that if the doctor or healthcare provider is the one doing the foot screening themselves, then it creates awareness within the professional as well too that for every patient I have to take care of their feet and not just look at their sugar level."* (S02P06, Family Physician)

## Referrals: interprofessional care amid systemic pressures

Referrals are a crucial part of the consultation as they ensure the continuum of care from the point of diagnosis to specialized care. In our study referrals to podiatrists were prolific (S4 Appendix). Notably, it was observed that referrals to

orthopaedics specialists were not made for cases with deformities, even though this action was recommended by the ACE guidelines. Furthermore, the results suggest that referrals to podiatrists extended beyond high-risk patients or those with active wounds or infections. Specifically, referrals to podiatrists were being issued for annual foot screenings, a service that can be alternatively, according to some study participants, provided by primary care doctors or nurses for low-risk patients.

*"Usually, I, because for me it's better doing an operation than not, because that's my job, right. So, if they have no wounds, um, they have no rest pain, that means they don't have pain in the foot when they're not even moving, um, there's no need for surgery. Er, so actually, we don't really do any, take wany, er, further investigations, or evaluation. Um, we just give them some general advice. Um, the podiatrist usually, I run the clinic with a podiatrist. So, the podiatrist will also give them some foot care and footwear advice. Um, and then we usually just refer them back to the Polyclinic."* (S04P04, Diabetes Nurse Educator)

The study participants have reported referrals to psychologists and social workers, which are brought up in the IWGDF as part of the assessment of person-related factors and are not encompassed within the ACE guideline. In certain cases, the referral process is optimized through a "one-stop-shop" concept where patients engage in consultations with various specialists all located within a single clinic. In other, participants underscore working together through leveraging their professional network.

*"Either we call the endocrinologist doctor [to] have a look or… we just drop by the next door the podiatrists there, ask them to do us a favour, come in to see the case."* (S01P03, Advanced Practitioner Nurse in Diabetes)

Post-referral appointments present challenges for both patients and HCPs. One concern is the access to healthcare services, including limited clinic hours and the distance. From the perspective of podiatrists, providing home visits may be a beneficial approach for patients with severe foot infections or limited mobility. This could also potentially help to identify environmental factors contributing to DFU (re-)occurrence or slow healing.

*"As I was saying just now, right, obviously, at the start, why are they calling the nurses to go to patients' homes when they have some medical issues, why not for diabetic foot? When, like I say, it's legs that they need to walk with, so it might be very difficult for them to travel from home to the centres or to the hospitals? So, that would be, yeah, that would be something to think about, yeah, in my opinion."* (S02P03, Advanced Practice Nurse)

Additionally, a high number of appointments may be burdensome, contributing to missed visits. Multidisciplinary centres may address this challenge by consolidating care. However, securing appointment slots can be difficult, causing delays. Regulations to limit wait times are not always effectively implemented. This is especially critical for DFU recurrence, where patients must restart the referral process. To avoid delays, some doctors admit patients for acute care, ensuring faster diagnostic scans but occupying beds needed for other acute cases.

*"Now, that can be a delay, and sometimes in order to overcome a delay we have to admit a patient in, take up an acute care bed, just to expedite the whole system of getting a scan, uh, which is not ideal."* (S03P03, Vascular surgeon)

### Wound care: combining therapeutic strategies and medical procedures

An integral component of wound care includes prescriptions and investigations, which encompass a range of pharmacologic or therapeutic interventions, such as anti-fungal creams, antibiotics to combat infections, glucose control therapies or prescription of foot-immobilizing or offloading devices.

*"We just prescribe them wound sandals, or, you know, those wound shoes which are much more suitable."* (S04P04, Vascular Surgeon)

Medical procedures are a crucial part of wound management and depend on the profession. In the case of podiatrists, it usually includes wound debridement, nail-cutting, dressing management, and wound cleaning including draining the infection. For vascular surgeons, it may comprise revascularization (either endovascular angioplasty, surgical bypass, or hybrid), skin grafting, amputation, endovascular angioplasty, or surgical bypass. Additionally, assessing financial and psychosocial aspects is important by evaluating if receiving subsidized consumables would support the patient in wound healing.

*"We'll do regular wound care, wound management, looking at the psychosocial as well, because you know… if they have any financial issues, then that's when we flag up the social workers so that they can… receive our consumables… that could help with their wound healing"* (S02P01, Senior Podiatrist)

## Discussion

This study provides insights into the real-world clinical practices of DFU prevention and care management in Singapore, assessed against the local and international DFU care model, presenting insights for healthcare policymakers and care pathway architects. o the best of our knowledge such inquiry is a unique example of such audit. The analysis revealed that clinical practice in the city-state is mostly aligned with and follows the main recommendations set forth IWGDF and ACE. During clinical visits, healthcare professionals perform foot assessments primarily through visual inspections and evaluations of vascular and neurological health. Education on DFU prevention and management is prevalent across all diabetes care facilities. Additionally, healthcare professionals exhibit a proactive and innovative approach to enhancing the impact of their work. However, there is still room for improvement as deviations from the recommended practices (underutilized risk stratification and DFU classification, tactile examination instead of monofilament or tuning fork), or contradicting recommendations occur (recommendations about Crocks). Our study suggests that there are several practices that clinical guidelines authors, healthcare system managers, and clinicians could consider to better streamline DFU prevention and care.

Firstly, it appears that foot screening and foot care education seem to be performed in a recurring manner across all DFU-related care and referrals to podiatrists extend beyond active wounds to include low-risk cases. While providing patient education on different stages of care pathway can be beneficial [19], repeated foot screening may result in unnecessary use of resources and may impact patient outcomes, experience and satisfaction. Our data suggests that almost all study participants (17/19) conducted at least one type of assessment when receiving a patient and that out of 12 non-podiatry HCPs, seven refer their patients to podiatrists, including standard annual foot screening, which may suggest that low-risk cases are undertaken excessively by podiatrists.

Notably, Jepson et al. report that more than half of podiatrists report a high frequency of screening of people with very low or moderate risk of diabetes-foot-related disease [20]. An audit study in Canada revealed that there are gaps in knowledge about the roles and scope of practice of disciplines involved in DFU [21]. One systematic review has discussed the risk of non-systematic, *ad hoc* patterns of patient referral in contrast with formalized protocol for referrals [22].

Subsequently, it appears to be an anticipation that primary care doctors will assume a greater role in patient screening and treatment to mitigate the volume of referrals to specialists. One of the study participants implemented foot screening as part of regular diabetes care emphasizing that this was not the usual routine. The Singapore Ministry of Health advocates earlier screening in the primary care setting [23] and there are already some initiatives of DFU multidisciplinary workflow in primary and tertiary care [13], including screenings conducted by specialized diabetes nurses in polyclinics

and subsidized primary care and, recently, adding risk stratification to electronic medical records. However, the local clinical care guideline, ACE, does not outline the expectation of primary care doctor's explicit and enhanced involvement. Similarly, IWDGF does not include specific recommendations on the primary care doctor's role in DFU prevention and management.

Considering the above, the delineation of roles, responsibilities and task delegation engaged in diabetic footcare should be more explicitly outlined in the guidelines. We acknowledge that the development of guidelines on role distribution in DFU prevention and management depends heavily on healthcare resources available in a specific country. It poses a challenge to draft them in a manner that maps the responsibilities of various healthcare actors and can be universally applicable across varying resource levels. Nevertheless, we believe it is imperative to undertake this endeavour.

Secondly, guidelines and hospital workflows, depending on the resources, could also put forward recommendations for home visits in severe cases and telemedicine in less critical cases. In contrast to IWGDF and ACE, international consensus on DFU in the Asia-Pacific region [9], includes a recommendation of telemedicine as an alternative to standard care and additionally suggests encouraging patients to monitor their wounds via photographs and a wound journal. Meloni et al. reported on implementing telemedicine during COVID-19 among patients with healed ulcers, uncomplicated DFUs, or in the case of those who suffer from 3 or more comorbidities [24]. Home visits on other hand, though time-consuming for HCPs, could alleviate the burden of commuting to the clinic when the patient is suffering from active DFU, hence reducing the risk of further progression. Jafary et al. report on an incremental cost-effectiveness ratio per quality-adjusted life year of home care treatment in comparison to hospital-based care and suggest that providing health care to patients with DFU at home is more cost-effective and could result in the reduction of mortality and readmission rates in addition to increased satisfaction of both patients and health care providers [25]. Cost-effectiveness may be, however, dependent on a specific country's resources and healthcare financing model.

Thirdly, the study participants also recognized the significance of integrating social welfare and psychological referral as part of the process of care. The IWGDF guidelines touch upon the need for assessing patients for depression and other psychosocial challenges [7,26]. In contrast, the ACE guidelines do not address psychosocial well-being at all. There is growing evidence for the beneficial influence of psychological support on DFU outcomes [27], which underscores the importance of placing greater emphasis on integrating provision of psychosocial support and exploring into clinical pathways.

## Limitations

This study offers valuable insights into clinical practices for preventing and managing DFUs in Singapore. However, it is important to acknowledge its limitations. The study comprised 19 participants recruited from five hospitals and healthcare institutions. To enhance the generalizability of the findings and solidify the observed patterns, it would be beneficial to expand the sample size. Furthermore, to gain a more comprehensive understanding of the overlaps and discrepancies in care provision across various clusters, i.e., primary care as well as private healthcare, further research is warranted. In particular, the perceptions of what other HCPs do, what types of assessments they conduct, and what is their referral decision tree, could discover possible misconceptions about the area of responsibility.

## Conclusions

While HCPs generally adhere to international and local clinical guidelines in DFU prevention and care management there is still room for improvement. Clinical guidelines architects, healthcare system managers, and clinicians should consider establishing clearer delineation of roles in DFU clinical care guidelines to improve the process of care. Clinical practice audits help to indicate areas in which guidelines can be enhanced which have the potential to impact DFU outcomes, decreased amputation rates, and significant cost savings.

## Supporting information

**S1 Appendix. SRQR.**
(PDF)

**S2 Appendix. FGD guide.**
(PDF)

**S3 Appendix. Codebook.**
(PDF)

**S4 Appendix. Visualization of the codes.**
(PDF)

## Acknowledgments

No artificial intelligence (AI)-assisted tools, chatbots or image creators were used for data analysis and interpretation.

## Author contributions

**Conceptualization:** Josip Car, James Best, Choon Huat Gerald Koh, Lorainne Tudor Car, Kavita Venkataraman, Bernhard Otto Boehm, Huiling Liew, Elaine Tan, Harvinder RS Sidhu, Tavintharan Subramaniam, Rosalind Chiew Jiat Siah, Andy Hau Yan Ho.

**Data curation:** Anita Pienkowska, Naren K Surendra, Kelley Goh, Nandika Lodh.

**Formal analysis:** Anita Pienkowska, Naren K Surendra, Kelley Goh, Nandika Lodh, Andy Hau Yan Ho.

**Funding acquisition:** Josip Car, James Best, Choon Huat Gerald Koh, Lorainne Tudor Car, Kavita Venkataraman, Bernhard Otto Boehm, Elaine Tan, Harvinder RS Sidhu, Rosalind Chiew Jiat Siah, Andy Hau Yan Ho.

**Investigation:** Anita Pienkowska, Josip Car, Huiling Liew, Elaine Tan, Harvinder RS Sidhu, Tavintharan Subramaniam, Naren K Surendra.

**Methodology:** Josip Car, James Best, Choon Huat Gerald Koh, Lorainne Tudor Car, Kavita Venkataraman, Bernhard Otto Boehm, Huiling Liew, Elaine Tan, Tavintharan Subramaniam, Andy Hau Yan Ho.

**Project administration:** Anita Pienkowska, Naren K Surendra.

**Resources:** Andy Hau Yan Ho.

**Supervision:** Josip Car, Andy Hau Yan Ho.

**Visualization:** Anita Pienkowska.

**Writing – original draft:** Anita Pienkowska, Kelley Goh, Nandika Lodh, Andy Hau Yan Ho.

**Writing – review & editing:** Anita Pienkowska, Josip Car, James Best, Choon Huat Gerald Koh, Lorainne Tudor Car, Kavita Venkataraman, Bernhard Otto Boehm, Huiling Liew, Elaine Tan, Harvinder RS Sidhu, Tavintharan Subramaniam, Rosalind Chiew Jiat Siah, Naren K Surendra, Andy Hau Yan Ho.

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
