## [Decision Letter · Decision Letter 0]

5 May 2025

Dear Dr. Ho,

Please revise the manuscript addressing all the comments from the two reviewersThe manuscript requires a minor revisionRemember to adhere to all formatting and styling requirements accordingly.

We look forward to receiving your revised manuscript.

Kind regards,

Fredirick Lazaro mashili, MD, PhD

Academic Editor

PLOS ONE

“This research is supported by the Singapore Ministry of Health’s National Medical Research Council under the Health Services Research Grant (HSRG-DB17Nov002).“

“This study was funded by the Singapore Ministry of Health’s National Medical Research Council under the Health Services Research Grant (HSRG-DB17Nov002), awarded to JC and AHYH. The funder nad no  role in the study design, data collection and analysis.”

3. In the online submission form, you indicated that [Deidentified interview data are available on request. Researchers who wish to obtain further details may contact the corresponding author with a methodologically sound proposal.].

4. Please amend your manuscript to include your abstract after the title page.

Additional Editor Comments:

Please revise and address all the comments from both the reviewers.

Reviewers' comments:

Reviewer's Responses to Questions

**Comments to the Author**

1. Is the manuscript technically sound, and do the data support the conclusions?

Reviewer #1: Yes

Reviewer #2: Yes

2. Has the statistical analysis been performed appropriately and rigorously?

Reviewer #1: Yes

Reviewer #2: Yes

3. Have the authors made all data underlying the findings in their manuscript fully available?

Reviewer #1: Yes

Reviewer #2: Yes

4. Is the manuscript presented in an intelligible fashion and written in standard English?

Reviewer #1: Yes

Reviewer #2: Yes

Reviewer #1: Abstract:

Well written, though a need to indicate on study design, further while you elaborated how you collected a brief indication of how participants were recruited, how you attained saturation and how transcription was done might improve the flow.

Introduction

Oky, as it has provide justification for the study

Methodology

Study design well written and justified

Add a brief rationale for selecting the specific healthcare clusters also consider clarifying how they data collector agreed upon saturation.

Additionally, as qualitative research rely on physical cues and expressions and you conducted interviews via zoom how were you able to capture these important aspect of the study

Results

Well written with a good flow of ideas

Expand just slightly on findings related to footwear education, what does it signify as most participant didn't mention it

Discussion

Okey. Though consider shortening slightly; some arguments (e.g., role delineation) are repeated. while noting out repeated screening having significant implication in the effectiveness of quality of care for both caregivers and patients you can enhance it by linking it with specific data from result as it points out a major weakness in service provision (fragment care)

Conclusion,

Its aligns with findings and discussion. But you also add actionable recommendations for guideline makers and policy makers

General comment

The paper is well written and significant in improving clinical practice specifically on the field of diabetic foot ulcer thus recommended for submission after comments being addressed

Reviewer #2: The results are presented in a structured and logically organized way, using categories that reflect the study’s framework — namely clinical guideline components. The findings are rich, well contextualized, and supported by participant quotes. However, I encourage the authors to refine their theme labels to reflect analytical insight, not just descriptive domains (suggestions, “Assessment: consistency and limitations in screening practice”, patient education: high value but varied depth, guidelines: selective application and gaps in utility”). Also, to more explicitly demonstrate how the framework guided the structure and interpretation of findings (Suggestion: In results or methods, briefly mention how each theme maps onto the clinical guideline components (e.g., ACE, IWGDF)—this helps anchor the analysis in the stated framework method). Additional participant quotations in later sections, particularly “Wound Care,” would strengthen the balance across themes (Suggestion: Add one or two direct quotes in wound care and referral sections — particularly where the discussion includes logistics, cost, or health systems barriers.

**Do you want your identity to be public for this peer review?** For information about this choice, including consent withdrawal, please see our Privacy Policy

Reviewer #1: No

Reviewer #2: **Yes: ** Fredirick mashili

---

## [Author Response · Author response to Decision Letter 1]

13 Jun 2025

Dear Reviewers,

Thank you for considering our manuscript and for taking the time to review it. Please find below answers to the review comments. Thank you for inviting us to revise the manuscript for your further consideration.

Response to Reviewers' comments:

Reviewer's Responses to Questions

Comments to the Author

1. Is the manuscript technically sound, and do the data support the conclusions?

Reviewer #1: Yes

Reviewer #2: Yes

=>We appreciate your feedback.

2. Has the statistical analysis been performed appropriately and rigorously?

Reviewer #1: Yes

Reviewer #2: Yes

=>We appreciate your feedback.

3. Have the authors made all data underlying the findings in their manuscript fully available?

Reviewer #1: Yes

Reviewer #2: Yes

=>We appreciate your feedback.

4. Is the manuscript presented in an intelligible fashion and written in standard English?

Reviewer #1: Yes

Reviewer #2: Yes

=>We appreciate your feedback.

5. Review Comments to the Author

Reviewer #1: Abstract:

Well written, though a need to indicate on study design, further while you elaborated how you collected a brief indication of how participants were recruited, how you attained saturation and how transcription was done might improve the flow.

=>Thank you for highlighting the need to improve the abstract’s flow. We have revised it to include clearer information on the study design, sampling and transcription process.

"A qualitative study design involving focus group discussions was used to explore practices in the prevention and treatment of DFUs. Verbatim transcripts from eight discussions involving 19 HCPs, purposively sampled, were analyzed using the framework method. Coding was guided by a DFU model of care developed from international and local clinical guidelines.

Introduction"

Oky, as it has provide justification for the study

=>Thank you for your kind and encouraging review.

Methodology

Study design well written and justified

Add a brief rationale for selecting the specific healthcare clusters also consider clarifying how they data collector agreed upon saturation.

Additionally, as qualitative research rely on physical cues and expressions and you conducted interviews via zoom how were you able to capture these important aspect of the study

=>Thank you for this helpful comment. We have added a brief rationale for the selection of specific healthcare clusters

"Participant recruitment was conducted at five major public hospitals and medical clinics in Singapore covering two out of three healthcare clusters. The initial study design included all three clusters to capture a broad spectrum of service settings and ensure diverse perspectives on DFU prevention and treatment. However, one cluster opted not to participate further due to differing institutional policies related to data sharing considerations for a connected sub-project."

and further clarified how data saturation was determined.

"Preliminary analysis of the verbatim transcripts was conducted during the data collection phase. Preliminary analysis of the verbatim transcripts was conducted concurrently with data collection. JC and AHYH engaged in iterative discussions and based on emerging patterns and recurring themes jointly concluded that data saturation had been reached after 19 participants were interviewed."

=>Regarding the use of Zoom for focus group discussions, we acknowledge that non-verbal cues can significantly enrich qualitative data. While non-verbal cues were not systematically recorded for analysis, the interviewers were attentive to participants’ expressions and reactions during the sessions and responded accordingly in real time. Given the nature of our research, which centers on clinical practices rather than emotional or behavioral nuances, we believe that the absence of formal non-verbal analysis does not significantly affect the robustness of our findings.

Results

Well written with a good flow of ideas

Expand just slightly on findings related to footwear education, what does it signify as most participant didn't mention it

=>Thank you for your helpful suggestion and supportive comments.

Regarding the findings on foot care education, we have expanded the section to include the number of participants who mentioned each topic, to illustrate the distribution of focus across different areas. We also clarified the statement concerning guideline-recommended topics that were not mentioned and discussed possible reasons for their omission.

"Footcare education provided by the study participants primarily focused on topics such as footwear (9/19 participants), including shoe shape and size or the use of socks; general diabetes education (8/19); wounds (8/19); inspection (4/19); hygiene (3/19); nails (3/19); and moisturizing (2/19). However, several specific recommendations outlined in clinical guidelines—such as inspecting shoes before wearing them, ensuring sole flexibility, using breathable materials, adjustable fastenings, or low heels—were not mentioned. This may suggest that these elements are either not perceived as immediately relevant in the context of this patient population or that there is a need to enhance the communication of guideline-based recommendations. Notably, we observed inconsistent recommendations about CrocsTM, alternating between endorsing and advising against their use."

Discussion

Okey. Though consider shortening slightly; some arguments (e.g., role delineation) are repeated. while noting out repeated screening having significant implication in the effectiveness of quality of care for both caregivers and patients you can enhance it by linking it with specific data from result as it points out a major weakness in service provision (fragment care)

=>Thank you for your careful review. We have adjusted the flow of the discussion section to minimize repetitions.

Conclusion,

Its aligns with findings and discussion. But you also add actionable recommendations for guideline makers and policy makers

=>We appreciate your supportive review.

General comment

The paper is well written and significant in improving clinical practice specifically on the field of diabetic foot ulcer thus recommended for submission after comments being addressed

=>Thank you for your supportive review. All comments have been addressed.

Reviewer #2: The results are presented in a structured and logically organized way, using categories that reflect the study’s framework — namely clinical guideline components. The findings are rich, well contextualized, and supported by participant quotes. However, I encourage the authors to refine their theme labels to reflect analytical insight, not just descriptive domains (suggestions, “Assessment: consistency and limitations in screening practice”, patient education: high value but varied depth, guidelines: selective application and gaps in utility”). Also, to more explicitly demonstrate how the framework guided the structure and interpretation of findings (Suggestion: In results or methods, briefly mention how each theme maps onto the clinical guideline components (e.g., ACE, IWGDF)—this helps anchor the analysis in the stated framework method). Additional participant quotations in later sections, particularly “Wound Care,” would strengthen the balance across themes (Suggestion: Add one or two direct quotes in wound care and referral sections — particularly where the discussion includes logistics, cost, or health systems barriers.

=>We appreciate your supportive review and are grateful for your insightful and helpful comments.

The subheadings in the findings section have been revised to reflect analytical insights rather than merely descriptive content. For the sections on assessment, patient education, and guidelines, we adopted your insightful suggestions: Assessment: consistency and limitations in screening practice, Patient education: high value but varied depth, and Guidelines: selective application and gaps in utility. For the remaining sections, we used the following titles: Referrals: interprofessional care amid systemic pressures and Wound care: combining therapeutic strategies and medical procedures.

Thank you for suggesting that we explicitly demonstrate how the framework guided the structure and interpretation of our findings. We have expanded the description of the model’s development and clarified how the data informed and shaped the final codebook.

"The research team (NKS and AP) developed a high-level model of care, outlining essential components visualized in Fig 1. Key guidelines such as ACE (10), IWGDF (6-8), and Diabetic Foot Ulcer Care in the Asia-Pacific region International Consensus Document (9) were iteratively analyzed to glean recurring and emphasized elements of DFU prevention and care. We began with iteratively drafting a decision flow diagram representing patient journey through different healthcare professionals, based on outcomes from various assessment, investigations and treatment strategies criteria (e.g. presence or absence of an active ulcer) drawn from the guidelines. To enhance clarity and simplify the diagram, we examined the main sections of each guideline to identify priority areas. The ACE guideline emphasizes foot assessment (including active DFU presentation, risk stratification, other contributing factors, and referrals) and patient education. The IWGDF guidelines cover pathophysiology, prevention, assessment and treatment, active Charcot neuro-osteoarthropathy, and organization of care. The Asia-Pacific consensus document highlights assessment, investigations, infection management, clinical treatment, and strategies for improving outcomes, such as multidisciplinary care, patient-centered approaches, self-care, telemedicine, and caregiver involvement. Based on this synthesis, we grouped the previously mapped steps into three interdependent domains—assessment, referral, and specialist care—with patient education as a cross-cutting element.

[…] The high-level model of care (Fig. 1) guided our iterative coding process. Insights from the FGD transcripts influenced its structure, resulting in a codebook which organizes data on “specialist care” under two categories: wound care and guidelines and offers more nuanced codes for patient education and referrals – the diverse information provided by participants impacted the granularity of codes for educational formats and topics, as well as additional codes for organizing care. In addition, we created a distinct category for guideline use and wound care practices across specialties (Fig. 2) to capture the declared adherence to clinical guidelines and the range of therapeutic and treatment strategies."

6. PLOS authors have the option to publish the peer review history of their article (what does this mean?). If published, this will include your full peer review and any attached files.

Yes

Regards,

Prof. Andy Ho

Anita Pienkowska

---

## [Decision Letter · Decision Letter 1]

4 Jul 2025

Real-world clinical practice of Diabetic Foot Ulcer prevention and care in Singapore: A qualitative inquiry with healthcare professionals

PONE-D-25-17596R1

Dear Dr. Ho,

We’re pleased to inform you that your manuscript has been judged scientifically suitable for publication and will be formally accepted for publication once it meets all outstanding technical requirements.

Kind regards,

Fredirick Lazaro mashili, MD, PhD

Academic Editor

PLOS ONE

Additional Editor Comments (optional):

All the previous comments and concerns have been sufficietly addressed

Reviewers' comments:

Reviewer's Responses to Questions

**Comments to the Author**

Reviewer #1: All comments have been addressed

Reviewer #2: All comments have been addressed

2. Is the manuscript technically sound, and do the data support the conclusions?

Reviewer #1: Yes

Reviewer #2: Yes

3. Has the statistical analysis been performed appropriately and rigorously?

Reviewer #1: Yes

Reviewer #2: Yes

4. Have the authors made all data underlying the findings in their manuscript fully available?

Reviewer #1: Yes

Reviewer #2: Yes

5. Is the manuscript presented in an intelligible fashion and written in standard English?

Reviewer #1: Yes

Reviewer #2: Yes

Reviewer #1: The study has been well written and add up t the body of knowledge linking up knowledge and practices through a well narrated design. Thus, i would offer t for publication

Reviewer #2: The authors have sufficiently addressed all the previous comments raised. This manuscript can now be accepted for publication

**Do you want your identity to be public for this peer review?** For information about this choice, including consent withdrawal, please see our Privacy Policy

Reviewer #1: No

Reviewer #2: **Yes: ** Fredirick mashili

---

## [Editor Report · Acceptance letter]

PONE-D-25-17596R1

PLOS ONE

Dear Dr. Ho,

I'm pleased to inform you that your manuscript has been deemed suitable for publication in PLOS ONE. Congratulations! Your manuscript is now being handed over to our production team.

Kind regards,

on behalf of

Dr. Fredirick Lazaro mashili

Academic Editor

PLOS ONE